# Cooperative colloidal self-assembly of metal-protein superlattice wires

Ville Liljeström[1,2], Ari Ora[2], Jukka Hassinen[1], Heikki T. Rekola[3], Nonappa[1], Maria Heilala [1], Ville Hynninen[1], Jussi J. Joensuu[4], Robin H.A. Ras [1,2], Päivi Törmä[3], Olli Ikkala [1] & Mauri A. Kostiainen[1,2]

Material properties depend critically on the packing and order of constituent units throughout length scales. Beyond classically explored molecular self-assembly, structure formation in the nanoparticle and colloidal length scales have recently been actively explored for new functions. Structure of colloidal assemblies depends strongly on the assembly process, and higher structural control can be reliably achieved only if the process is deterministic. Here we show that self-assembly of cationic spherical metal nanoparticles and anionic rod-like viruses yields well-defined binary superlattice wires. The superlattice structures are explained by a cooperative assembly pathway that proceeds in a zipper-like manner after nucleation. Curiously, the formed superstructure shows right-handed helical twisting due to the right-handed structure of the virus. This leads to structure-dependent chiral plasmonic function of the material. The work highlights the importance of well-defined colloidal units when pursuing unforeseen and complex assemblies.

[1] HYBER Centre of Excellence, Department of Applied Physics, Aalto University, FI-00076 Aalto, Finland. [2] HYBER Centre of Excellence, Department of Bioproducts and Biosystems, Aalto University, FI-00076 Aalto, Finland. [3] COMP Centre of Excellence, Department of Applied Physics, Aalto University, FI-00076 Aalto, Finland. [4] HYBER Centre of Excellence, VTT Technical Research Centre of Finland Ltd, 02150 Espoo, Finland. Correspondence and requests for materials should be addressed to M.A.K. (email: mauri.kostiainen@aalto.fi)

upramolecular self-assembly provides a method to control structural features from the sub-nanometer to the nanometer level. At longer length scales, supracolloidal self-assembly[1, 2] has been introduced to allow additional levels of hierarchy, making the construction of diverse structures and functions possible[3–7]. It is well established that for example shape[8], charge[4, 9], specific nearest neighbor interactions[10, 11], and patchiness[12] can be used to control the arrangement of colloidal nanoparticles into higher order structures[5]. Controlling the assembly conditions and interaction strengths is particularly important. The assembly kinetics play a significant role in achieving well-defined structures, as rapid self-assembly can yield kinetically trapped particle configurations even for spherical particles[4]. To extend the concept of supracolloidal self-assembly to include an increasing number of practical applications, the methods need to be robust and yield desired structures reproducibly, that is, to be deterministic[12, 13].

A self-assembly pathway can be deterministic and yield a higher degree of order if the subunits assemble in a cooperative manner. In the context of supramolecular polymerization, cooperative assembly allows non-covalently connected subunits to form ordered structures instead of randomly ordered aggregates[14]. Another example shows that cooperative complexation of polyelectrolyte-surfactant systems leads to well-defined self-assembled periodicities, independent of the exact polyelectrolyte-surfactant composition[15]. Cooperative self-assembly dominated by nucleation and growth can be also utilized on the supracolloidal

level. The assembly could result in hierarchical periodic structures, even though the colloidal constituents have a considerably lower mobility than in the polymer-surfactant system. In addition, the supracolloidal cooperative self-assembly can lead to superstructures even at nanoparticle stoichiometries, which could assumedly lead to a complete lack of order.

Here the cooperative self-assembly is demonstrated in a colloidal system consisting of anionic tobacco mosaic virus (TMV) nanorods, which direct the assembly of cationic gold nanoparticles (AuNPs) (Fig. 1). Nanoparticles can generally exhibit size-dependent magnetic and plasmonic properties, and they have been used to form superlattices or other constructions that have properties arising from the superstructure[16–18]. Furthermore, there are a number of sophisticated examples that show how chiral colloidal particles can be used to guide the formation of finite nanoparticle assemblies with structure dependent optical properties[19–21]. In this study, the combination of building units forms processable macroscopic wires with structure dependent optical properties.

## Results

**Charged colloidal particles**. In this work we used AuNPs with a narrow size distribution ($D_{core} = 12.4 \pm 0.9$ nm), which were functionalized with a covalently linked (11-Mercaptoundecyl)-$N$, $N,N$-trimethylammonium bromide (MUTAB) ligand giving the AuNPs a hydrodynamic radius of $\sim 15$ nm, a highly cationic

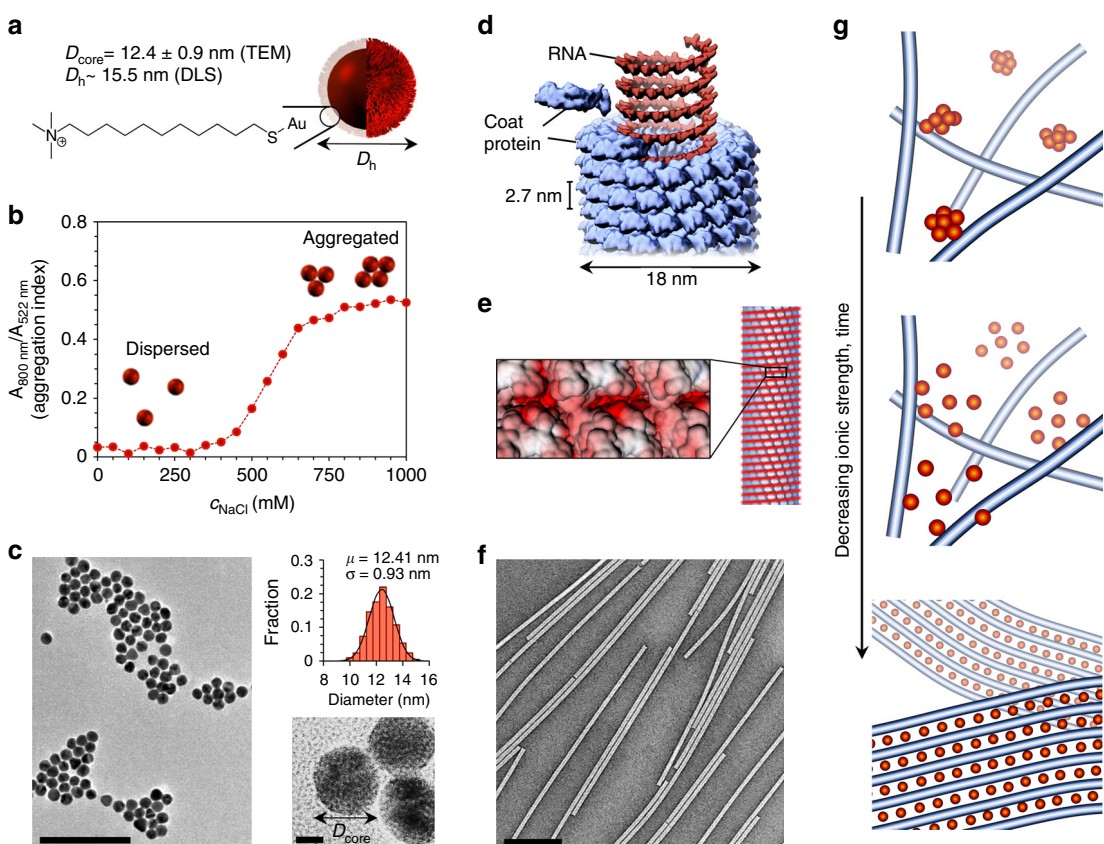

**Fig. 1** Structural details of the oppositely charged colloids and their electrostatic self-assembly upon dialysis. **a** Details of the used gold nanoparticles and chemical structure of the cationic ligand. **b** Colloidal stability of the AuNPs shown as the aggregation index, which is a function of the ionic strength, characterized by UV–vis spectrometry. The aggregation index is defined here as $A_{800\,nm}/A_{522\,nm}$. **c** TEM image of AuNPs and the AuNP size distribution from 205 particles. **d** Schematic of the structural details of TMV. **e** The electrostatic surface potential map of TMV in aqueous solution shows that TMV surface charge distribution has a helical symmetry. **f** TEM micrograph of the TMV nanorods. **g** Schematic illustration of the formation of the binary assemblies as a function of the ionic strength. *Scale bars*: 100 nm and 5 nm in **c** and 200 nm in **f**

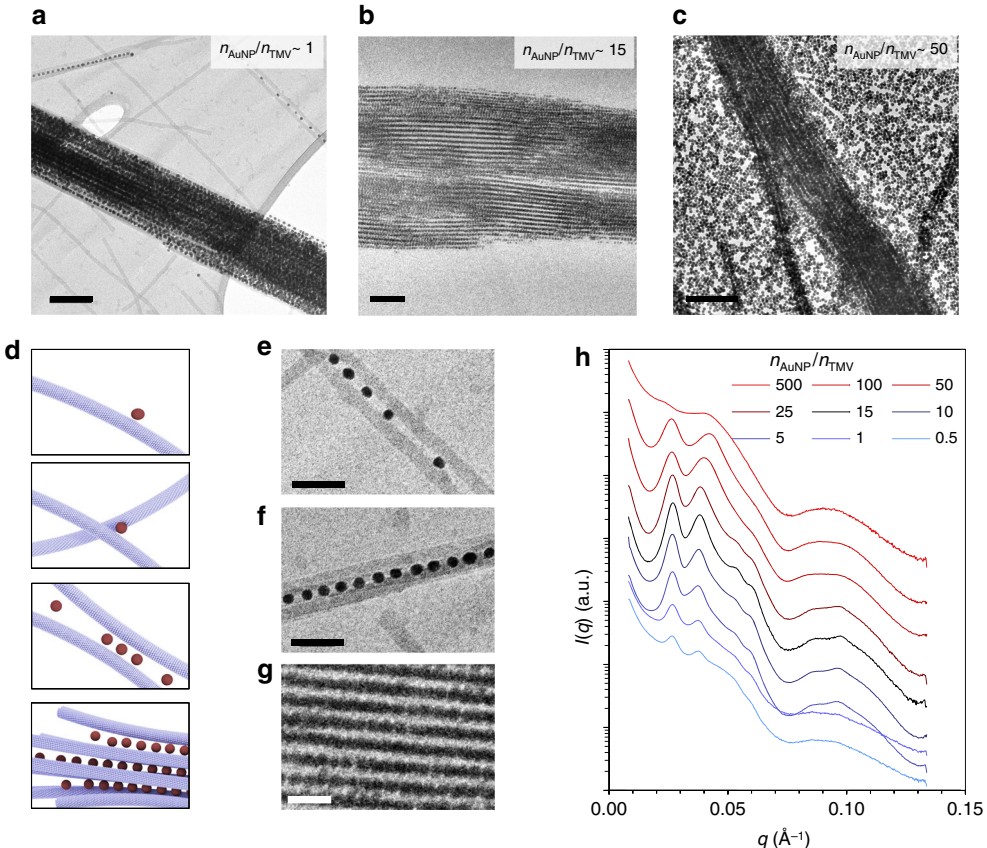

**Fig. 2** Cooperative assembly yields well-ordered superlattice structures. **a** Cryo-TEM image of TMV-AuNP structures in a sample with an excess of TMV. **b** Cryo-TEM image of TMV-AuNP structures in a sample with optimal stoichiometric ratio of the particles. **c** Conventional TEM image of TMV-AuNP structures in a sample with an excess of AuNPs. **d** Schematic illustration of the cooperative self-assembly process where nucleation of the assembly occurs, when two TMVs are crosslinked by one (or more) AuNP, forming a focused attractive potential leading to the formation of nanoparticle rows. **e**, **f**, Close-up of details in **a**, showing AuNP rows formed between two (**e**) and four (**f**) TMVs. **g** Close-up of a detail in **b**, showing a well-ordered TMV-AuNP superlattice. **h** SAXS data measured from samples with different particle stoichiometries (indicated in the legend). *Scale bars*: 200 nm in **a**–**c**. 50 nm in **e**–**g**

surface, and a high colloidal stability at a range of ionic strengths (Fig. 1a–c). Native TMV (Fig. 1d–f), a semiflexible rod-like colloid with a diameter of 18 nm and length of 300 nm, is a protein cage consisting of 2130 coat proteins, which are helically wrapped around the viral RNA[22]. The helical arrangement of the coat proteins is also prevalent in the electrostatic surface potential map, which forms a negatively charged helix on the surface of the TMV. TMV is a versatile bioparticle that can be diversely modified[23] and has been widely used as a model platform in material science, for example, when probing electrolyte theories[24] or guiding the assembly of small functional molecules[25, 26] or inorganic materials[27, 28]. TMV has also proven to have a substantial affinity to multivalent metal cations leading to the formation of ordered assemblies[29]. Combining the properties of TMV and AuNP in a superlattice makes an interesting hybrid material. Forming superlattice structures is especially interesting as the asymmetric helical and exactly defined structure of TMV can transfer chirality into higher order structures[30]. The previously reported template based chiral nanoparticle assemblies have not yet demonstrated clear periodic three-dimensional (3D) structural order of multiple components.

**Cooperative electrostatic self-assembly.** As oppositely charged particles easily form kinetically trapped structures at low ionic strengths, we chose to mix the AuNPs and TMV at such a high ionic strength ($c_{NaCl} = 500$ mM) that the electrostatic attraction

between them is efficiently screened and the like-charged AuNPs are significantly aggregated. We then sequentially dialyzed the mixture against stepwise decreasing ionic strength, which reinstalls efficient electrostatic attraction and causes the AuNPs to form assemblies with TMV (Fig. 1g).

To understand the formation of binary assemblies, the effect of particle stoichiometry $n_{AuNP}/n_{TMV}$ was studied. A total failure of the assembly of highly ordered structures could also be assumed in conditions, where the $n_{AuNP}/n_{TMV}$ is either too low or too high. Therefore we prepared a sample series with $n_{AuNP}/n_{TMV}$ covering three orders of magnitude ranging from 0.5 to 500. All studied $n_{AuNP}/n_{TMV}$ yielded assemblies that were visually observed as a violet precipitate. Unbound AuNPs resulted in a ruby red supernatant in samples containing an excess of AuNPs. Samples including an excess of TMV had a clear or no supernatant and had a gel-like appearance (Supplementary Fig. 1). To gain more insight into the structure of the assemblies, we characterized the samples using small-angle X-ray scattering (SAXS). Unexpectedly, clear diffraction peaks were identified from the SAXS profiles of almost all the measured samples (Fig. 2h), which indicates highly ordered superlattice structures. The best resolved diffraction peaks were found in samples with $n_{AuNP}/n_{TMV}$ ~ 10–25, whereas less distinctive peaks were observed outside this optimal range of $n_{AuNP}/n_{TMV}$. At $n_{AuNP}/n_{TMV}$ ~ 500, the diffraction peaks were almost completely washed out, suggesting that the formation of the superlattices can ultimately be prohibited by an extreme excess of AuNPs. However, the first

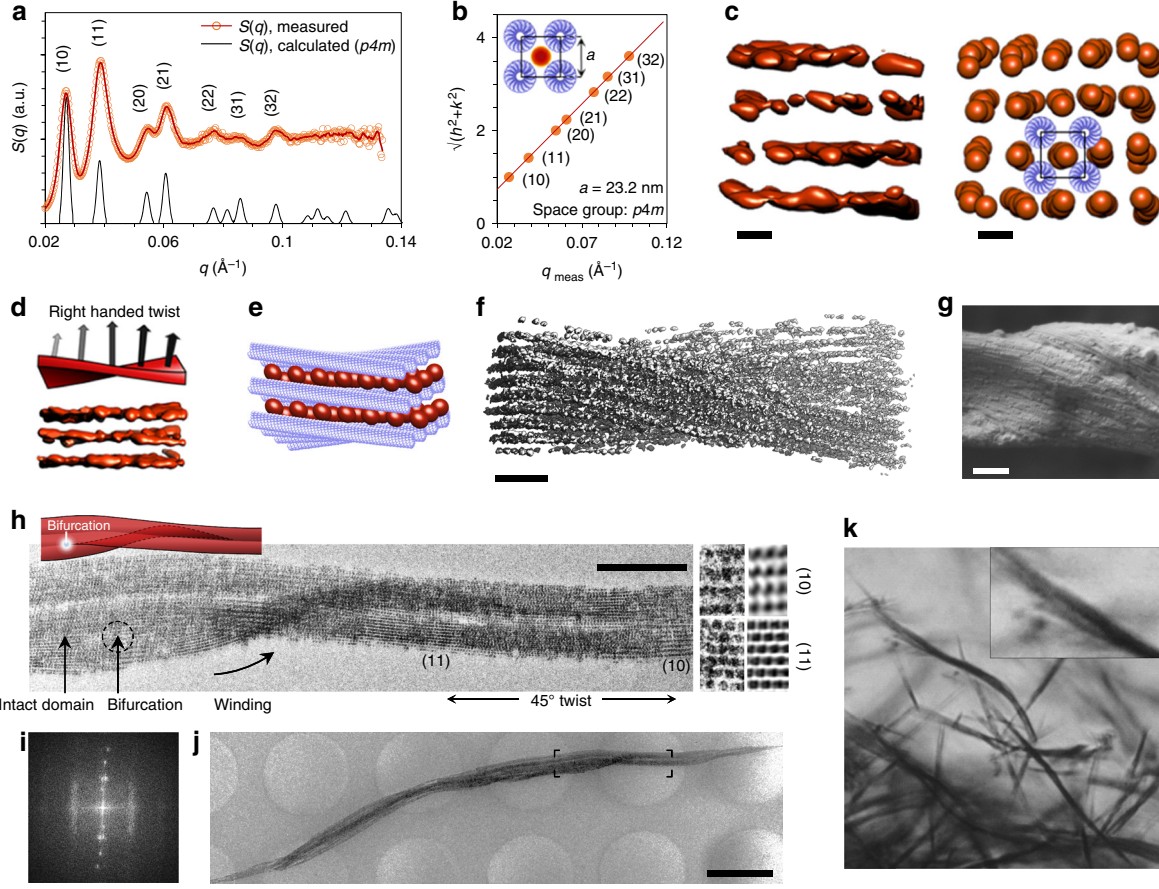

**Fig. 3** Hierarchical structure of the superlattice wires. **a** Measured and calculated structure factor $S(q)$ ($n_{AuNP}/n_{TMV}$ ~ 15). $a = 23.2$ nm is assumed in the calculation. **b** The squared diffraction peak indices plotted as a function of measured $q$-values. A linear fit yields $a = 23.2$ nm. **c** A cryo-ET density map (*left*) of a superlattice wire cross section ($n_{AuNP}/n_{TMV} \approx 15$). The red spheres (*right*) denote AuNPs that are positioned at local density maxima (*right*) and form a twisting 2D square lattice with the indicated TMVs. **d** 3D density isosurface of a selected volume of the cryo-ET reconstruction showing the lattice twist on a nanoparticle level. **e** A schematic of the superlattice including TMVs. **f** Cryo-ET reconstruction of a single superlattice wire. **g** SEM image of a plunge-frozen and freeze-dried superlattice wire. **h** Cryo-TEM image (magnification of **j**) of several aligned superlattice domains that wind around each other. Close-up shows (10) and (11) lattice planes. A bifurcating domain is indicated with an arrow. **i** FFT of **h** shows maxima for periods of 25 nm, 16.5 nm, 10.5 nm, 8.3 nm, which correspond to (10), (11), (21), and (31) lattice plane distances (vertical direction) and maxima that refer to periods of 16.2 nm and 8.1 nm, which resemble an interparticle distance of 16.2 nm for AuNPs in the aligned rows (horizontal direction). **j** A low magnification image of a bundle of superlattice wires. **k** An optical microscope image of superlattice fibers ($n_{AuNP}/n_{TMV}$ ~ 10). The close-up highlights winding of fibers. *Scale bars*: 20 nm **c**, 100 nm **f**, 200 nm **g**, 500 nm **h**, 2 μm **j**, 50 μm **k**

significant conclusion drawn from the SAXS data is that ordered superstructures are prone to form regardless of the $n_{AuNP}/n_{TMV}$, which suggests that the assembly mechanism strongly directs the assembly formation after nucleation. AuNP-TMV superlattices were also seen in both conventional transmission electron microscopy (TEM) and cryogenic electron microscopy (cryo-TEM) images. In all the studied samples, the superlattices had a fibrillary morphology. Samples with a relative excess of TMV ($n_{AuNP}/n_{TMV}$ ~ 1) contained unbound TMVs, whereas all the AuNPs were bound into highly ordered AuNP-TMV super-structures or between two or more TMVs forming intermediate zipper-like configurations (Fig. 2a). Significantly, it shows that like-charged AuNPs crowd the interstitial space between adjacent TMVs rather than smoothly covering all the TMVs. At optimal $n_{AuNP}/n_{TMV}$ ~ 15, large superlattices are formed and no free TMVs or AuNPs are observed (Fig. 2b). Furthermore, TEM images of samples with an excess of AuNPs ($n_{AuNP}/n_{TMV}$ ~ 50) still showed similar superlattice structures (Fig. 2c), even though a significant amount of free AuNPs was present.

The features observed in the SAXS and TEM data can be fully explained in terms of cooperative assembly (Fig. 2d): superlattices

nucleate when TMVs are crosslinked by AuNPs. The crosslinked complex attracts AuNPs to a higher degree than single TMVs. The crosslinked TMVs become aligned, and the nearest free AuNPs are subject to a focused attractive field between the TMVs. Therefore, the interstitial channel between TMVs is energetically a highly favorable position for AuNPs, leading to a zipper-like growth (Fig. 2e), where nanoparticles crosslink TMVs into bundles (Fig. 2f), forming large superlattice structures (Fig. 2g). Although additional TMVs are incorporated both in the TMV axis and the normal-to-axis direction, the zipper-like assembly mechanism with the rod-shape of TMV provides fast growth in the TMV axis direction, leading to a high aspect ratio of the formed superlattices. Furthermore, observing the assembly in situ using optical microscopy reveals the spontaneous release of AuNPs and the instantaneous formation of needle-like filaments when decreasing ionic strength (Supplementary Fig. 2).

**Hierarchical structure of the superlattice.** For rods, two-dimensional (2D) hexagonal close packing is typical, as it is the most efficient form of close packing, and it has been reported

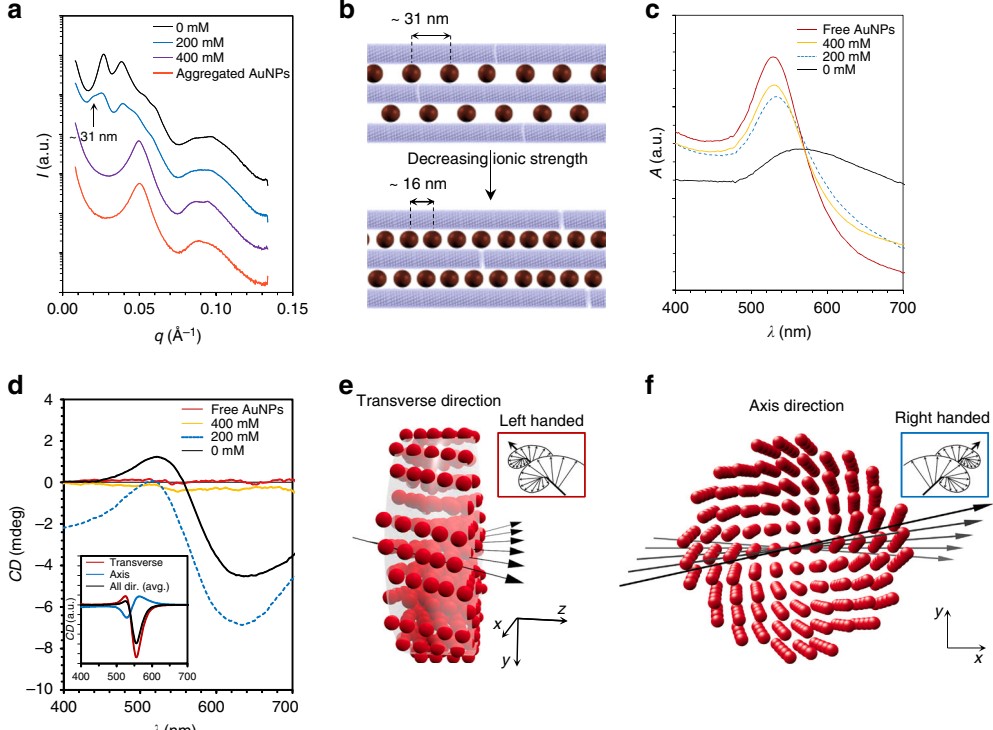

**Fig. 4** Optical properties as a function of the superstructure. **a** SAXS data showing the formation of the superlattices upon decreasing ionic strength ($n_{AuNP}/n_{TMV}$ ~ 25 for all measured data). Data shifted in $y$-direction for clarity. The *arrow* indicates the diffraction peak ($q$ ~ 0.02 Å$^{-1}$) corresponding to the interparticle distance ($d$ ~ 31 nm) of nanoparticles within individual rows of nanoparticles. **b** Upon decreasing ionic strength, AuNPs are concentrated into the superlattice wires leading to a smaller AuNP interparticle distance. **c** UV–vis data (scaled for clarity) at different salt concentrations showing the effect of AuNPs assembling tightly in the superlattice as a function of the ionic strength. The plasmonic resonance peak at 524 nm corresponding to free AuNPs disappears completely only at low ionic strength. **d** CD data (Savitzky–Golay filtered for clarity, original data in Supplementary Fig. 12). CD features appear as the superlattice forms upon decreasing ionic strength. Inset shows the simulated CD spectrum for different directions of light propagation and taken as an average of all directions. **e**, **f** The generated nanoparticle model including 400 AuNPs used to model the CD spectrum. The figure indicates that the lattice undergoes a left-handed **e** or a right-handed **f** twist depending on the viewing direction

that at high concentrations TMV forms a 2D hexagonal lattice in inclusion bodies found in infected cells[31] or when confined by polyelectrolytes[32]. The analysis of the SAXS data showed that here the TMVs also arrange into a 2D array, though contrary to previous observations the structure factor $S(q)$ equaled that of a 2D square lattice (space group = $p4m$, Fig. 3a, b) with a lattice constant of 23.15 nm (in excellent agreement with close packing of the building blocks). However, the typical 2D hexagonal close packed lattice was observed when mixing TMVs and strongly cationic particles smaller than the AuNPs described here (Supplementary Fig. 3). In conclusion, the large size of the AuNP allows it to bind to four TMVs and still form a close packed 2D square lattice. Smaller nanoparticles cannot efficiently bind to four TMVs, yielding a hexagonal lattice, where each nanoparticle is bound to three TMVs. Furthermore, from the SAXS results and the $n_{AuNP}/n_{TMV}$ ratios, we conclude that statistically the average distance between adjacent AuNPs in highly ordered superlattices is ~ 15–30 nm.

In TEM sample preparation, the superlattice wires settle on the TEM grid perpendicular to the viewing direction, thus inhibiting the direct observation of the cross section of superlattice wires, and hence the 2D square lattice. Therefore, we conducted cryogenic electron tomography (cryo-ET) reconstruction of the samples to verify and amplify our understanding of the superlattice structure. The 2D square lattice was evident from the cryo-ET 3D reconstruction (Fig. 3c, Supplementary Movies 1 and 2). Further-more, the 3D reconstruction clearly showed that the superlattice wires have a helical structure (Fig. 3d–f), evident even on

nanoparticle level and seen as a right-handed axial twisting of the (10) lattice planes. From the cryo-ET, the twist $\omega$ (360°/helical pitch) was estimated to be ~ 0.13°/nm at its steepest. A similar right-handed helical structure of the superlattice wires is also observed in scanning electron microscopy (SEM) images (Fig. 3g), from which $\omega$ is estimated as ~ 0.08°/nm at its steepest. The magnitude of $\omega$ can also be estimated directly from plain TEM images. The lattice vectors and lattice planes rotate about the rotation axis as a function of the position on the rotation axis in a helical superlattice. Consequently, even though the superlattice is continuous, any lattice directions that are identified from the transmission micrographs appear as finite sized domains with a period attributed to the lattice plane distance, and any observable lattice directions are shown at regular intervals (Supplementary Fig. 4). For example in Fig. 3h, (10) and (11) lattice planes can be identified from the lattice periods that are estimated from the micrograph as 25 nm ($\approx$ lattice constant $a$) and 17 nm ($\approx a/\sqrt{2}$) respectively (23.2 nm and 16.4 nm according to SAXS data). The distance between the observed (10) and (11) directions is roughly 1.5 µm, corresponding to $\omega$ ~ 0.03°/nm as the angle between (10) and (11) planes is 45°. From fast Fourier transform (FFT) analysis, the periods corresponding to the (21) and (31) lattice planes are also identified, and in total the FFT analysis yields a lattice constant of 24.9 nm. Observing smaller $\omega$ for larger superlattices suggests that the initial nanoparticle level interaction leads to helicity of the superlattice, but $\omega$ is reduced upon superlattice growth.

The location of AuNPs relative to TMV in the superlattice axis direction is not particularly constrained, but the distance between

adjacent AuNPs, in the same interstitial space, is relatively constant (Figs. 3i and 4a). The interparticle distance is controlled by the competing repulsive (AuNP–AuNP) and attractive (AuNP–TMV) interactions, which are not strong enough to facilitate the formation of a 3D lattice. However, for a true 3D lattice, a helical twisting of a periodic structure, as observed here, is forbidden, as this would break the translational symmetry in the direction of the rotation axis.

The growth of helical nanoparticle wires with a 2D cross-sectional lattice is explained by an assembly mechanism, which assumes that AuNPs are dynamically moving in the interstitial space. In this model, the right-handed helical twist can be explained in terms of electrostatic interaction between a positively charged 0D object and a negative helical charge distribution; the helical superstructure is a net effect of the strong electrostatic attraction between AuNPs and TMV, and the helical symmetry of TMV (Supplementary Figs. 5–7 and Supplementary Discussion). Helicity initiated from nanoparticle level interactions is also indicated by cryo-TEM, which shows winding of TMVs around a row of AuNPs (Fig. 2f).

As seen in Figs. 2 and 3, the size of intact helical superlattice domains can grow large. The TMVs are mainly aligned in the rotation axis direction, but the peripheral TMV nanorods within a superlattice are increasingly tilted as a function of the superlattice width, which is evident from the tomography and SEM data (Fig. 3f, g). Internal geometrical frustration affects the structure of filaments consisting of linear subunits[33]. Here, we also argue that the helical structure forms a constraint for the radial growth of helical superlattice wires, even though $\omega$ is relatively modest. This is supported by three observations. First, filaments with smaller width have larger $\omega$ (Fig. 3f) compared with thick filaments (Fig. 3g, h). Second, large domains apparently bifurcate (Fig. 3h); and third, thick fibers consist of bundled superlattice wires instead of large single domain superlattices (Fig. 3j, k).

**Emergent chiral optical properties**. The cooperative assembly of superlattice wires is a way to form macroscopic chiral plasmonic structures. This was demonstrated by conducting both SAXS and circular dichroism (CD) measurements of the water-borne superlattice wires at different stages of the assembly. From the SAXS studies (Fig. 4a and Supplementary Fig. 8) we confirmed that superlattice formation is a continuous process during which the AuNPs are able to diffuse in the interstitial channels within the superlattice structures, thus allowing the AuNPs to concentrate into the superstructures upon decreasing ionic strength. Importantly, the interparticle distance between adjacent AuNPs is large at high ionic strength (Fig. 4a, b and Supplementary Fig. 8f) and decreases as the ionic strength decreases, allowing a high AuNP content in the superlattices. The formation of helical plasmonic superlattice wires is clearly observed in the CD spectra at visible wavelengths, which proves that the helical twist has a preferred handedness instead of being a racemic mixture[34]. The components do not show any circular dichroism at a high ionic strength where assembly of superlattice structures has not yet occurred. Upon assembly, the baseline decreases below zero mdeg, and a peak-dip feature, similar to those that are often related to helical plasmonic structures, is observed[19]. Upon further decreasing the ionic strength, the baseline goes to zero, and the spectral features shift slightly. However, the peak-dip shape of the spectra remains. From our experiments, we conclude that the characteristics of the CD spectra are directly related to the AuNP superstructures, especially as the CD spectra could be partly washed out by simple mechanical shear forces that break the superstructures (Supplementary Fig. 9).

Coupled dipole approximation simulations provided a coherent description of the CD spectra[35]. The simulation was based on 400 AuNPs arranged into a helical superlattice structure with a lattice constant of 23.15 nm and an interparticle distance of $16 \pm 1.6$ nm. A right-handed helical structure ($\omega = 0.13°/$nm) was assumed (Fig. 4e, f). The simulation reproduced the main features of the CD-spectrum (Fig. 4d, inset). The mismatch in the width and position of the peak-dip feature of measured and simulated data is mostly explained by a variation of the width and $\omega$ of the superlattices within real samples (see Supplementary Figs. 10 and 11 for CD modeling of different structures). Importantly, for such superlattice structures, the simulations revealed that the CD is highly dependent on the orientation of the structure, as this kind of a finite superlattice undergoes both right-handed (axis direction) and left-handed (transverse direction) twists depending on the viewing direction. Here the left-handed directions dominate in the simulated average CD spectrum. This is also the case in the CD experiment, as the typical length of a superlattice wire is in micron-scale, which leads to almost complete absorption of light propagating in the axis direction.

Furthermore, the plasmonic superlattice wires were found to be strongly polarizing. We demonstrated in a proof-of-concept manner that such structures can be used to prepare plasmonic polarizers. AuNP-TMV superlattice wires with a slight excess of TMV ($n_{AuNP}/n_{TMV} \sim 10$) were further functionalized with cationic magnetic $Fe_3O_4$ nanoparticles (Fig. 5a). The mixture contained a slight excess of $Fe_3O_4$ nanoparticles so that the superlattice wires were saturated with $Fe_3O_4$ nanoparticles. Functionalized superlattice wires were characterized by energy dispersive X-ray spectroscopy (EDX) mapping (Fig. 5b), which confirmed a high iron content for superlattice wires functionalized with cationic $Fe_3O_4$ nanoparticles. Such waterborne structures could be aligned in a magnetic field and dried into a film that exhibits both plasmonic and polarizing properties (Fig. 5c–e). Observing the film through crossed polarizers confirmed the global alignment of the superlattice wires. Rotating the film in an angle $\theta$ showed that the transmitted intensity $I(\theta)$ was directly proportional to $(\sin\theta \cdot \cos\theta)^2 + $ constant. For a perfect linear polarizer $I(\theta)$ would be directly proportional to $(\sin\theta \cdot \cos\theta)^2$. Functionalization by $Fe_3O_4$ nanoparticles induces partial aggregation of superlattice wires, which explains the deviation from an ideal polarizer (Supplementary Fig. 13). Aggregated superlattice wires are kinetically locked and not able to align in the magnetic field but give rise to an isotropic background. This proof-of-concept showed that hierarchical self-assembly approaches can be utilized to produce new materials with tuneable physical properties.

## Discussion

We have described the cooperative self-assembly mechanism that is involved in the formation of binary electrostatic self-assemblies of anionic 1D and cationic 0D colloidal particles. Due to the cooperative zipper-like assembly, superlattice wires form in a wide range of nanoparticle stoichiometries. The typical structure formed by close-packed monodisperse nanorods is 2D hexagonal lattice. On the contrary, we show that a superlattice consisting of these particles has 2D square lattice geometry. This results from the strong electrostatic attraction between the oppositely charged particles and the particle size ratio, which allows each AuNP to bind to four TMVs.

Self-assembly of exactly defined objects have previously demonstrated unexpected and highly sophisticated structures. This has a great impact on the development of new materials. A continuous development of functional nanomaterials requires efficient methods to organize functional components into desired

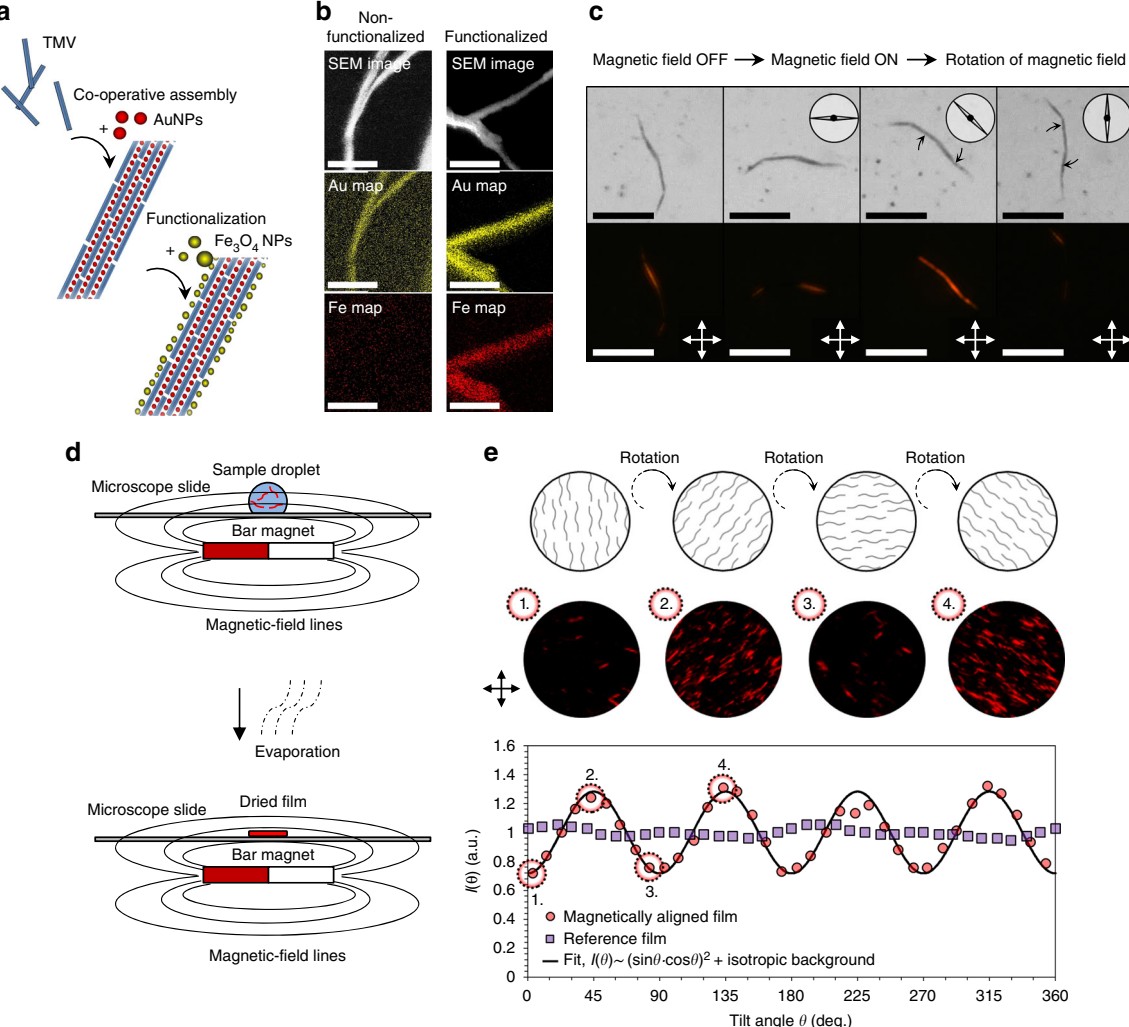

**Fig. 5** Magnetic alignment of functionalized superlattice wires yields a plasmonic polarizer. **a** Scheme of the preparation of magnetic superlattice wires. **b** SEM images and EDX maps of non-functionalized and functionalized superlattice wires. Functionalized superlattice wires show a smooth coverage of iron implying that $Fe_3O_4$ nanoparticles bind efficiently to the wires. **c** Optical (*above*) and polarized light (*below*) microscope images of a functionalized waterborne superlattice wire, which is rotated by a magnetic field. The direction of the magnetic field is indicated by the schematic compass needle. Segments of the superlattice wire that are oriented in the directions of the crossed polarizers are not visible. **d** An oriented film was prepared by drying a suspension of $Fe_3O_4$ nanoparticle functionalized superlattice wires in a magnetic field (5 ± 1 mT). **e** The linear polarization of the film was demonstrated by polarized light microscopy. The transmitted intensity $I(\theta)$ can be modulated by rotating the film in an angle $\theta$ with respect to the crossed polarizers as structures oriented in the directions of the crossed polarizers are not visible. Four examples of images (the diameter of the circular area is 0.3 mm) that were used to obtain $I(\theta)$ data points are shown. The reference film was formed without an applied magnetic field. The direction of the crossed polarizers are indicated by double arrows **c**, **e**. *Scale bars*: 2.5 μm **b** and 50 μm **c**

assemblies. We observe an unforeseen helical superlattice structure, which originates from the nontrivial and complex electrostatic interaction between the TMV with a helical charge distribution and the oppositely charged AuNP.

In the AuNP-TMV superlattice wires the helical superstructure is efficiently combined with plasmonic interaction with light. Importantly, the helical organization of the nanoparticles results in plasmonic circular dichroism at visible wavelengths. On a general level this result exemplifies that structural properties of superlattices can be combined with specific properties of individual nanoparticles (physical, chemical, or other properties) yielding collective properties that exceeds the properties of the individual nanoparticles. This requires, however, that both the nanoscale structural features and the colloidal interactions are sufficiently well controlled. It can be expected that the advances in the field of designed protein cages[36–38] will yield colloidal particles with a controlled chirality and shape, which could enable

precise control over the handedness, macroscopic habit, and physical properties of multicomponent superlattices.

Here, the geometry of the nanoparticles and the ionic-strength-controlled interparticle interaction leads to a distinct wire-like habit of the superlattices. In addition, we show that functionalizing the superlattice wires with magnetic nanoparticles enables a global control of the anisotropic optical properties exhibited by the superlattice wires. In principle the wire-like habit of the structures also enables industrial scale roll-to-roll fiber processing of hierarchically structured materials. The described self-assembly approach utilizes exactly defined colloidal units and addresses the need for screening new self-assemblies that are required for future materials.

## Methods

**Preparation and characterization of tobacco mosaic virus particles**. A month-old *Nicotiana tabacum* (var. Samson) plants grown in greenhouse were inoculated by rubbing the leaves with wild type TMV stock. After 22 days from inoculation,

the upper leaves showing the most severe TMV symptoms were collected from six plants. TMV purification was made starting from 52 g of leaf biomass according to Chapman[39]. Briefly, the leaves were homogenized in 0.5 M phosphate buffer (pH 7.2) with 1% (v/v) 2-mercaptoethanol. Filtered leaf juice was extracted with butan-1-ol, and following centrifugation, the aqueous phase was collected. PEG 8000 was used in a concentration of 4% to precipitate the TMV. The virus pellet was washed with ice cold 10 mM phosphate buffer (pH 7.2) containing 0.7 M NaCl and 4% PEG and finally resuspended.

The TMV was further purified by sucrose gradient centrifugation. The TMV band was collected and finally dialyzed three times against 10 mM sodium acetate buffer (pH 5.5, dialysis time: 3 h, 3 h, 24 h). The TMV concentration and the particle composition were characterized using UV–visible spectrophotometry (UV–vis) and TEM. The absorbance at 260 nm and 280 nm was measured from a dilution series (20 ×, 40 ×, 80 ×, and 120 × dilution) using PerkinElmer LAMBDA 950 UV/Vis spectrophotometer and a quartz cuvette. At 260 nm TMV has an extinction coefficient of $3.0 (cm\ l^{-1})^{-1}$ i.e. 118200 cm$^{-1}$ mM$^{-1}$ [40]. On the basis of the UV–vis results the TMV concentration in the starting solution was 34.5 mg ml$^{-1}$, i.e., 0.876 μM. The $A_{260}/A_{280}$ ratio $R$ was measured as 1.03, which indicates a slight excess of protein ($R_{CP} = 0.65$[41], $R_{RNA} = 2$[41] and $R_{TMV} = 1.2$[42]). In TEM intact TMV particles with a length of 300 nm or longer were observed. The coat protein is known to form single helices at acidic (pH < 6) conditions, which may extend the TMV particles[43].

**AuNP synthesis**. The synthesis of cationic AuNPs followed mainly the AuNP synthesis described by Hassinen et al.[44]. Here the synthesis is described briefly and more details of the synthesis can be found elsewhere[44, 45]. In total 500 ml of 0.5 mM HAuCl$_4$ dissolved in water was boiled. Around 50 ml of preheated 38.8 mM Na$_3$Ct (trisodium citrate tribasic dehydrate, ≥ 99.0%) was then poured into the boiling solution under vigorous magnetic stirring. The color of the initially yellow solution turned immediately clear. 1 min later the solution turned black and after 7 min the solution was dark ruby red. Heating was stopped after 14 min and the AuNP solution was cooled down at room temperature.

**AuNP cationization**. In the first phase transfer AuNPs were conjugated with oleylamine (OLAM) and transferred into an organic solvent (CHCl$_3$) as follows. 275 ml of the citrate stabilized AuNP solution was poured into a separatory funnel. First the pH was adjusted by adding 2.75 ml 0.1 M NaOH. Then 13.75 ml of CHCl$_3$ and 688 μl OLAM (70% purity) was added and the mixture was shaken vigorously after which the CHCl$_3$ phase was let to separate. The CHCl$_3$ phase was then collected and the residual AuNPs were washed from the remaining aqueous phase by adding 3 ml of CHCl$_3$. The mixture was again shaken vigorously. The red CHCl$_3$ phase was again collected. This was repeated three times after which the aqueous phase was transparent. The collected volume was centrifuged to further separate any residual water from the solution.

In the second phase transfer AuNPs were conjugated with (11-mercaptoundecyl)-N,N,N-trimethylammonium bromide (MUTAB) and transferred back to aqueous phase as follows. 4.4 ml H$_2$O and 550 μl of 100 mM MUTAB (dissolved in ethanol) were added to the collected CHCl$_3$ phase. The mixture was shaken resulting in a foamy emulsion, which was dissolved by adding 110 μl of 1 M HCl and shaking. The sample mixture was then centrifuged to separate the aqueous phase. After centrifugation the mixture consisted of a clear colorless CHCl$_3$ phase (bottom), a foamy emulsion (middle), and a dark ruby red aqueous phase (uppermost). The uppermost phase was collected and washed twice with 5 ml of CHCl$_3$ (addition of CHCl$_3$, shaking, centrifuging, and collecting the aqueous phase) to remove residual OLAM. Finally the MUTAB conjugated AuNPs were dialyzed three times against milliQ water to remove excess MUTAB and to neutralize the pH.

The AuNP mass concentration in the stock solution was measured by lyophilizing a known volume of AuNP stock solution and weighing the mass, whereas the AuNP core-size distribution was measured using TEM (see below), and the hydrodynamic diameter $D_H$ was measured using DLS (see below). The AuNP number density $n_{AuNP}$ was calculated from the known mass concentration, by assuming a spherical shape for the AuNP, and an average MUTAB surface density of 5.6 MUTAB/nm$^2$. Because of the errors in AuNP size, shape, and MUTAB surface density, the calculated $n_{AuNP}$ are considered to have an accuracy of ± 10%.

**Synthesis of cationic magnetic iron oxide nanoparticles**. The iron oxide (Fe$_3$O$_4$) nanoparticles were prepared according to a modified procedure reported by An et al[46]. In short, 0.86 g of Fe(II)Cl$_2$ • 4H$_2$O and 2.35 g of Fe(III)Cl$_3$ • 6H$_2$O (in 1: 2 molar ratio) were dissolved in 40 ml of degassed MQ H$_2$O. 3.4 ml of 28–30% NH$_4$OH was added dropwise into the solution under vigorous stirring (400 rpm) and N$_2$ degassing. Black precipitate formed instantly. The particles were washed with three cycles of magnetic decantation followed by redispersion in degassed MQ H$_2$O. The particles were cationized by coating with (2,3-epoxypropyl)-trimethylammonium chloride (EPTMAC). For the EPTMAC functionalization, the magnetic decantation was repeated once more and the particles were redispersed in 60 ml of degassed 1.75 M NaOH. 20 ml of the nanoparticle solution was used to prepare the cationic Fe$_3$O$_4$ particles used in this study. The cationization was carried out according to a modified procedure reported by Hasani et al.[47]

EPTMAC was added to obtain roughly 1:1 molar ratio of iron and EPTMAC. The vials were filled with N$_2$, sealed and placed in an oil bath at 65 °C for 4 h under stirring. The vials were removed from the oil and washed with three cycles of magnetic decantation followed by redispersion in degassed MQ H$_2$O to remove unreacted EPTMAC. The pH of the mixtures was lowered to 3.0 with HCl and the vials sonicated for 2 min. Centrifugation (5000 RCF) for 3 min was used to separate and discard large aggregates. The supernatant was collected and characterized by DLS (Zetasizer Nano Series, Malvern Instruments). The mode size (volume fraction) was 58.9 nm and the zeta potential was at +43.6 mV.

**AuNP colloidal stability**. The AuNP colloidal stability upon increasing ionic strength was characterized by the aggregation index ($A_{800\ nm}/A_{522\ nm}$) as measured using UV-vis spectrophotometry (PerkinElmer LAMBDA 950 UV/Vis spectrophotometer). 1 M NaCl solution was sequentially added to 500 μl of dilute (maximum absorbance <2) AuNP solution to obtain the different NaCl concentrations. After each titration step the sample was mixed properly after which the UV-vis spectrum was measured. After reaching 250 mM NaCl concentration the ionic strength was increased by adding 5 M NaCl solution instead of 1 M NaCl solution to avoid excessive dilution of the sample.

**Dynamic light scattering**. The hydrodynamic diameter $D_H$ of the AuNP was measured using a standard dynamic light scattering device (Zetasizer Nano Series, Malvern Instruments) with a 4 mW He-Ne ion laser at a wavelength of 633 nm and an avalanche photodiode detector at an angle of 173°. Experiments were carried out at 25 °C. Plastibrand semi-micro PMMA cuvettes were used in the experiment and Zetasizer software (Malvern Instruments) was used to obtain the particle size distributions.

**Transmission electron microscopy**. Conventional transmission electron microscopy (TEM) imaging of TMV particles, AuNPs and AuNP–TMV superlattices was carried out with JEM-2800 High Throughput transmission electron microscope (JEOL) using an acceleration voltage of 200 kV. 3 μl of the sample was added on 200 mesh copper grid with carbon support film (CF-Quantifoil) and the excess sample was blotted away with filter paper. AuNP samples and AuNP–TMV superlattices were imaged without staining, whereas ammonium molybdate negative staining was used in imaging of the TMV particles.

**Cryogenic transmission electron microscopy and electron tomography**. The Cryogenic transmission electron microscopy (Cryo-TEM) images were collected using JEM 3200FSC field emission microscope (JEOL) operated at 300 kV in bright field mode with Omega-type Zero-loss energy filter. The images were acquired with Gatan Digital Micrograph software while the specimen temperature was maintained at −187 °C. The Cryo-TEM samples were prepared by placing 3 μl aqueous dispersion of the sample on a 200 mesh copper grid with holey carbon support film (CF-Quantifoil) and plunge freezed using vitrobot with 3 s blotting time under 100% humidity.

Electron tomographic tilt series were acquired with the SerialEM-software package[48]. Samples were tilted between ± 69° angles with 2–3° increment steps. Prealignment of tilt image series, fine alignment and cropping was executed with IMOD[49]. The images were binned 2–4 times to reduce noise and computation time. Maximum entropy method (MEM)[50] reconstruction scheme was carried out with custom made program on Mac or Linux cluster with regularization parameter value of $\lambda = 0.001$.

**Sample preparation for SAXS and cryo-TEM**. The superlattices were formed during dialysis in floating dialysis cups (MWCO 14 kDa, cellulose membrane) that floated on the dialysate, which consisted of 10 mM sodium acetate buffer (pH 5.5) and a specific NaCl concentration $c_{NaCl}$. The AuNP and TMV ($c_{NaCl} = 500$ mM for both) were mixed and set into the dialysis cup. The $c_{NaCl}$ for the dialysate was 500 mM at start and the $c_{NaCl}$ was decreased by 50 mM every 30 min until the desired $c_{NaCl}$ was reached. After reaching the final $c_{NaCl}$ the dialysate was changed one more time and the dialysis was continued for 2 h.

The sample for conventional TEM ($n_{AuNP}/n_{TMV} = 50$) was prepared by mixing AuNP and TMV at 100 mM NaCl concentration without further dialysis.

**Small-angle X-ray scattering**. The aqueous samples were sealed between two Kapton foils. The sample environment was evacuated to reduce background scattering from air. The small-angle X-ray scattering (SAXS) was measured using a Bruker Microstar microfocus rotating anode X-ray source (Cu Kα radiation, $\lambda = 1.54$ Å). The beam was monochromated and focused by a Montel multilayer focusing monochromator (Incoatec). The X-ray beam was further collimated by four collimation slits (JJ X-ray) resulting in a final spot size of less than 1 mm at the sample position. A Hi-Star 2D area detector (Bruker) was used to collect the scattered intensity. Sample-to-detector distance was 1.59 m, and a silver behenate standard sample was used for the calibration of the length of the scattering vector $q$. One-dimensional SAXS data was obtained by azimuthally averaging the 2D scattering data and the magnitude of the scattering vector $q$ is given by $q = 4\pi \sin\theta/\lambda$, where $2\theta$ is the scattering angle. The measured structure

factor $S(q)$ was obtained by dividing the scattering intensity $I(q)$ by the $I(q)$ measured from dissolved AuNPs according to the local monodisperse approximation[51]. The theoretical $S(q)$ was calculated with PowderCell[52].

**Scanning electron microscopy and energy dispersive X-ray spectroscopy.** Sample preparation for scanning electron microscopy (SEM) and energy dispersive X-ray spectroscopy (EDX) included liquid propane plunge freezing and subsequent freeze drying of suspensions deposited on TEM grids. SEM imaging was carried out using Zeiss Sigma VP electron microscope and EDX mapping was carried out using JEOL JSM-7500FA analytical field emission scanning electron microscope.

**Circular dichroism spectrometry and UV-visible spectrophotometry of AuNP-TMV assemblies.** The circular dichroism spectrometry (CD) measurements were carried out using a Chirascan (AppliedPhotophysics) spectrometer and a high quality quartz cuvette with a 10 mm light path (Hellma QS). The UV-visible spectrophotometry (UV-vis) data was measured simultaneously. The samples were prepared by dialysis according to the description above. All the samples contained in total 62 µg ml$^{-1}$ of AuNP and 3.9 µg ml$^{-1}$ of TMV except for the reference AuNP sample which contained 31 µg ml$^{-1}$ of AuNP. The samples were measured multiple times using a short measurement time and the sample was gently stirred between measurements to avoid the effect of sedimentation. The final data is an average of at least five short measurements.

**Modeling the circular dichroism.** Simulations of the circular dichroism (CD) signal are done with a coupled-dipole approach[35]. Each nanoparticle is modeled as a 12.5 nm diameter gold sphere, which interacts with an external, circularly polarized electromagnetic field and the fields produced by the other nanoparticles in the system. Quasistatic approximation is used to obtain the polarizability of an individual nanoparticle

$$\alpha_m = \alpha_m^3 \frac{\varepsilon_{\mathrm{NP},m} - \varepsilon_s}{\varepsilon_{\mathrm{NP},m} + 2\varepsilon_s}, \tag{1}$$

where $\alpha_m$ is the radius of $m$:th nanoparticle, and $\varepsilon_s$=1.8 is the relative permittivity of the surrounding media. For the relative permittivity of the nanoparticle $\varepsilon_{\mathrm{NP},m}$ we use tabulated data[53]. The dipole moment induced in a single nanoparticle is given by

$$\mathbf{d}_m = \alpha_m[\mathbf{E}_{\mathrm{ext}}(\mathbf{r}_m) + \mathbf{E}_{\mathrm{d-d}}(\mathbf{r}_m)], \tag{2}$$

here $\mathbf{E}_{\mathrm{ext}}(\mathbf{r}_m) = \mathbf{e}_0 \exp(i\sqrt{\varepsilon_s}\mathbf{k} \cdot \mathbf{r}_m)$ is the external electric field at the $m$:th particle and $E_{\mathrm{d-d}}$ is the electric field due to all other dipoles in the system

$$\mathbf{E}_{\mathrm{d-d}} = \sum_{m \neq n} \left[ \frac{3(\mathbf{d}_n \cdot \mathbf{n}_{mn})\mathbf{n}_{mn} - \mathbf{d}_n}{|\mathbf{r}_{mn}|^3}(1 - i\sqrt{\varepsilon_s}k|\mathbf{r}_{mn}|) + \frac{\varepsilon_s k^2(\mathbf{d}_n - (\mathbf{d}_n \cdot \mathbf{n}_{mn})\mathbf{n}_{mn})}{|\mathbf{r}_{mn}|} \right]. \tag{3}$$

In equation (3) $\mathbf{r}_{mn} = \mathbf{r}_m - \mathbf{r}_n$, and $\mathbf{n}_{mn} = \mathbf{r}_{mn}/|\mathbf{r}_{mn}|$. Eqs. (2) and (3) can be written in matrix form as $\mathbf{d} = \alpha\mathbf{E}_{\mathrm{ext}} + \beta\mathbf{d}$, where $\mathbf{E}_{\mathrm{ext}}$ now contains the external electric fields at each particle location, and the matrix $\beta$ contains the dipole-dipole interactions between the particles. The solution for the dipole moments is $\mathbf{d} = (1 - \beta)^{-1}\alpha\mathbf{E}_{\mathrm{ext}}$. We solve the matrix inversion numerically. Once the inverted matrix $(1 - \beta)^{-1}$ is known, the response of the structure to circularly polarized ($\pm$) external fields can be evaluated. The absorption cross-section for the nanoparticle assembly is

$$\sigma_{\pm} = 4\pi\sqrt{\varepsilon_s}k\,\mathrm{Im}\left[\sum_m \frac{\mathbf{d}_{m,\pm}^* \cdot \mathbf{d}_{m,\pm}}{\alpha_m^*}\right]. \tag{4}$$

The CD signal is then obtained by calculating the difference in the absorption cross-sections for different circular polarizations, averaged over solid angle $\Omega$ as the structures are free to move in the solution:

$$\Delta\sigma_{\mathrm{CD}} = \langle \sigma_+ - \sigma_- \rangle_{\Omega}. \tag{5}$$

**Data availability.** The authors declare that all data supporting the findings of this study are available from the corresponding authors on request.

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

## Acknowledgements

Jari Valkonen and Yan-Ping Tian are acknowledged for supplying the TMV used in the inoculation and for assisting in the purification of the TMV. Mikko Poutanen and Mika Latikka are acknowledged for valuable comments and insight. Financial support from the Academy of Finland (Grants 263504, 267497, 273645, 284621), the European Research Council (Grant No. ERC-2013-AdG-340748-CODE), Biocentrum Helsinki, and Emil Aaltonen Foundation and the Swedish Cultural Foundation in Finland are gratefully acknowledged. This work was carried out under the Academy of Finland's Centers of Excellence Programme and made use of the Aalto University Nanomicroscopy Centre (Aalto NMC).

## Author contributions

V.L., A.O., and M.A.K. formulated the project. V.L., A.O., J.H., and M.H. synthesized the AuNPs. J.J. prepared TMV. V.L., A.O., and M.H. performed conventional transmission electron microscopy imaging. V.L. and M.H. collected the UV–vis data. V.L., A.O., and J.H. collected the CD data. V.L. collected and analyzed the SAXS and SEM data as well as carried out the optical microscopy experiments. Nonappa performed cryo-TEM imaging. H.R. and V.L. planned and H.R. conducted the computational modeling of the CD signal. V.H. synthesized the magnetic nanoparticles. M.A.K. supervised the work. V.L. drafted the manuscript and O.I. and M.A.K. revised it. All authors commented on the manuscript.

## Additional information

**Competing interests:** The authors declare no competing financial interests.

