## [Peer Review File · Nature Communications]

Reviewers' comments:

Reviewer #1 (Remarks to the Author):

1. Who will be interested in reading the paper, and why?

Topics in self-assembly, nanotechnology, and protein engineering are covered in this paper. Thus, it will be of interest to researchers working in these areas.

2. What are the main claims of the paper and how significant are they?

This paper uses the tobacco mosaic virus (TMV) as a template for assembling cationic gold nanoparticles (AuNPs) into superlattice wires via electrostatic interactions. Charged AuNPs assemble cooperatively into the interstitial space between TMV fibers, forming a 2D square lattice that exhibits a right-handed fibrillar helical twisting. To avoid forming kinetically trapped structures, the researchers slowly increased the electrostatic attraction between AuNPs and TMV using a stepwise dialysis approach. Rather than the 2D hexagonal lattice normally expected, the researchers found that a 2D square lattice geometry could be assembled using larger AuNPs. Cryo-ET revealed that the 2D lattices exhibit a right-handed helical twist. The chiral optical properties of these structures were investigated using CD and corroborated with coupled dipole approximation simulations.

As cited in the paper, TMV is well-studied and has been used previously to deposit and organize metal, silica, and other types of nanoparticles into nanowires,¹⁻⁵ including chiral structures.⁶ Previous work has shown that a biomolecular scaffold can be used to template the assembly of cationic AuNP chains via electrostatic interactions to create higher order structures. Thus, the novelty of this work relies on the high quality of the ordered structures achieved and the thoroughness of characterization. The use of monodisperse AuNPs and a slow, electrostatic-mediated assembly to reduce kinetically trapped structures enables the synthesis of much higher quality superlattices than previously achieved. However, neither the right-handed helical nature nor 2D square lattice packing of the particles are entirely surprising given the nature of the viral template.⁷⁻⁸ As it stands, this paper is an elegant extension of existing work. Additional experiments could significantly increase the novelty however (detailed below).

3. Is the paper likely to be one of the five most significant papers published in the discipline this year?

Although the findings are not so unexpected, it is possible that the characterization and modeling methods used in the paper will make it a reference in the field.

4. How does the paper stand out from others in its field?

The excellent characterization, using a combination of SAXS, Cryo-ET, and the modeling of expected CD data (supplementary) make this paper stand out.

5. Are the claims novel? If not, which published papers compromise novelty?

In the context of this well-studied system, the novelty is limited to the improved quality of the assembled structure and excellence of the characterization rather than the findings themselves.

6. Are the claims convincing? If not, what further evidence is needed?

Yes, the claims are supported from the SAXS, Cryo-ET, and CD characterization.

7. Are there other experiments or work that would strengthen the paper further?

The novelty of the paper could be improved if more phases were shown to be accessible using this technique. The researchers show that the relative size of the AuNPs influences the symmetry of the resulting lattice, studying two sizes of AuNPs that access 2D square lattices and 2D hcp lattices. What is the effect of using AuNPs larger than 12 nm? Is there another phase that can be accessed? If

multiple chiral structures/phase transitions could be achieved by changing the relative size of the AuNP to the TMV, the novelty of the work would increase significantly and then this work would be appropriate for publication in Nature Communications.

Also, if magnetic particles were used (or a mixture of magnetic and plasmonic), the packing and helicity of the structure could be changed by altering the magnetic field. Such a system, which would be responsive to external stimuli, would be quite unique and interesting.

8. Questions for the Authors

- A. The SAXS peaks appear to shift to lower q as the ratio of AuNP:TMV is increased (Figure 2h), indicating that the spacing in the structure is increasing. Why is this the case?
- B. Have different cationic ligands been tried?
- C. What is the average length of the wires? Does this change at all with NP ratio (or other factors)?

9. Clarity of the manuscript

The manuscript is clearly written and organized. See detailed comments below.

[Page 3, line 7] this is an extremely long, run on sentence

[Page 3, line 11] What does "delicate" mean in this context?

[Page 3, line 14] TMV has been used to assemble chiral NP assemblies in the past and "exquisite" is quite a subjective term. Perhaps this could be revised to a more specific comment about the improvements made.

[Page 3, line 18] The phrase "at the onset of aggregation" is unclear and could be replaced with "dispersed."

[Page 5, line 12] "This is essentially one of the key findings" should be removed. You could say, "Importantly/ Significantly, it shows that like-charged..." in the next sentence instead, to make the same point.

[Page 7, line 20] "continueous" is spelled incorrectly and should be changed to "continuous"

References

1. Bromley, K. M.; Patil, A. J.; Perriman, A. W.; Stubbs, G.; Mann, S. *Journal of Materials Chemistry* 2008, 18, (40), 4796-4801.
2. Dujardin, E.; Peet, C.; Stubbs, G.; Culver, J. N.; Mann, S. *Nano Letters* 2003, 3, (3), 413-417.
3. Tseng, R. J.; Tsai, C.; Ma, L.; Ouyang, J.; Ozkan, C. S.; Yang, Y. *Nature nanotechnology* 2006, 1, (1), 72-77.
4. Fowler, C. E.; Shenton, W.; Stubbs, G.; Mann, S. *Advanced Materials* 2001, 13, (16), 1266-1269.
5. Lee, S.-Y.; Royston, E.; Culver, J. N.; Harris, M. T. *Nanotechnology* 2005, 16, (7), S435.
6. Kobayashi, M.; Tomita, S.; Sawada, K.; Shiba, K.; Yanagi, H.; Yamashita, I.; Uraoka, Y. *Optics express* 2012, 20, (22), 24856-24863.
7. Warner, M. G.; Hutchison, J. E. *Nature materials* 2003, 2, (4), 272-277.
8. Kostianen, M. A.; Hiekkataipale, P.; Laiho, A.; Lemieux, V.; Seitsonen, J.; Ruokolainen, J.; Ceci, P. *Nature Nanotechnology* 2013, 8, (1), 52-56.

Reviewer #2 (Remarks to the Author):

This article describes the co-assembly of 12 nm gold particles with tobacco mosaic virus (TMV) into higher ordered structures. Apparently, the gold particles can bridge up to four TMVs, which leads to the formation of chains of particles lying between the TMVs and thus forming ribbons of TMV-particle hybrids with micrometer diameter and hundreds of micrometer in length. By varying the respective concentrations of the the two constituents or of the ionic strength of the buffer solution, the authors

reveal details of the process of formation. The growth is explained by cooperative assembly and zipper-like closure of the particles between the TMVs leading to the formation of large superlattice structures. Further, the intrinsic chirality of the TMVs is passed on to the superstructures leading to twisting of the ribbons and thus measurable CD signals of the curled gold particle chains.

While I think this is a well written manuscript describing an interesting effect, I am undecided to recommend publication in Nature Communications in its current state. Although the authors have characterized the emerging structures rigorously, the story would benefit from a clearer purpose. Currently it appears to be of the sort "Let's see what happens if we mix A with B". E.g. if the hypothesized effect of the components assembling in a zipper-like fashion could be observed dynamically or could be modelled in silico there might evolve more general principles. Alternatively, the authors could rewrite the conclusion and instead of mainly repeating the content of the paper elaborate more on the "future materials" they have in mind.

Minor comments:

While the scheme in Fig 1g makes sense to me (decreasing ionic strength allow aggregates to dissolve and formation of bonds with the TMVs, I have trouble understanding the statement: "Upon decreasing ionic strength, AuNPs are concentrated into the superlattice wires leading to a smaller AuNP interparticle distance." A bit more explanation of the rationale behind that statement would be helpful.

"... the asymmetric helical and exactly defined structure of TMV can transfer chirality into the higher order structures." into the higher order of these structures? or ... into higher ordered structures?

REVIEWERS' COMMENTS:

Reviewer #1 (Remarks to the Author):

The novelty of the work is better stated and defended by the introduction of magnetic nanoparticles into the system to produce films of oriented superlattice wires, which demonstrate properties of a plasmon polarizer. In addition, upon reconsideration the descriptions of the helical nature of the superlattice via the bending and torsion mechanism are unique and informative. Furthermore, the formation of well-ordered 3D chiral structures allows for improved correlation of optical properties with structure.

However, in addition to elaborating on the practical applications, it would be valuable to discuss how these findings impact the field from a scientific perspective in the conclusion. For example, how would one use the information that you have learned in this work to inform the rational design of other types of chiral materials? Or, could one envision that the bending and torsion model might be applicable to other systems? In other words—beyond it being an interesting phenomenon, what implications does this work have for the field of self-assembly? If these minor concerns can be addressed, I believe the manuscript would be suitable for publication in Nature Communications.

Reviewer #2 (Remarks to the Author):

The authors have addressed the comments of the referees. The data of SI Figs 13 and 14 (magnetic induced polarization switching) are very impressive and I think they deserve space as a Figure 5 in the main text. Generally, the article has improved significantly and is now ready for publication.

Response to Reviewers' comments:

Reviewer #1 (Remarks to the Author):

1. Who will be interested in reading the paper, and why?

Topics in self-assembly, nanotechnology, and protein engineering are covered in this paper. Thus, it will be of interest to researchers working in these areas.

2. What are the main claims of the paper and how significant are they?

This paper uses the tobacco mosaic virus (TMV) as a template for assembling cationic gold nanoparticles (AuNPs) into superlattice wires via electrostatic interactions. Charged AuNPs assemble cooperatively into the interstitial space between TMV fibers, forming a 2D square lattice that exhibits a right-handed fibrillar helical twisting. To avoid forming kinetically trapped structures, the researchers slowly increased the electrostatic attraction between AuNPs and TMV using a stepwise dialysis approach. Rather than the 2D hexagonal lattice normally expected, the researchers found that a 2D square lattice geometry could be assembled using larger AuNPs. Cryo-ET revealed that the 2D lattices exhibit a right-handed helical twist. The chiral optical properties of these structures were investigated using CD and corroborated with coupled dipole approximation simulations.

Many thanks for the detailed comments and careful evaluation of our manuscript.

As cited in the paper, TMV is well-studied and has been used previously to deposit and organize metal, silica, and other types of nanoparticles into nanowires,¹⁻⁵ including chiral structures.⁶ Previous work has shown that a biomolecular scaffold can be used to template the assembly of cationic AuNP chains via electrostatic interactions to create higher order structures. Thus, the novelty of this work relies on the high quality of the ordered structures achieved and the thoroughness of characterization. The use of monodisperse AuNPs and a slow, electrostatic-mediated assembly to reduce kinetically trapped structures enables the synthesis of much higher quality superlattices than previously achieved. However, neither the right-handed helical nature nor 2D square lattice packing of the particles are entirely surprising given the nature of the viral template.⁷⁻⁸ As it stands, this paper is an elegant extension of existing work. Additional experiments could significantly increase the novelty however (detailed below).

We thank the reviewer for this comment and are happy that the work was found to be elegant. In our opinion the work has significant novelty. We have outlined the principles to obtain 2D lattices with unprecedented order and given a plausible mechanism for their cooperative formation. Furthermore, we see that the helical nature of the superlattice is both surprising and elegantly explained by the simultaneous bending and torsion mechanism. We have also made significant efforts to improve the manuscript (e.g. demonstrate magnetic field responsive structures) and improve novelty (see detailed comments below).

3. Is the paper likely to be one of the five most significant papers published in the discipline this year?

Although the findings are not so unexpected, it is possible that the characterization and modeling methods used in the paper will make it a reference in the field.

4. How does the paper stand out from others in its field?

The excellent characterization, using a combination of SAXS, Cryo-ET, and the modeling of expected CD data (supplementary) make this paper stand out.

Thank you for the supportive comment.

5. Are the claims novel? If not, which published papers compromise novelty?

In the context of this well-studied system, the novelty is limited to the improved quality of the assembled structure and excellence of the characterization rather than the findings themselves.

We thank the reviewer for this comment and we have made efforts to improve the novelty of the work by including a proof-of-concept application of the structures. See comment 7. We would also like to point out the obtained structures (helical superlattice wires) have not been reported before and therefore the novelty is not limited to characterization. We have also already carefully (and extensively) explored the formation mechanism and explained why the chiral structures are formed.

6. Are the claims convincing? If not, what further evidence is needed?

Yes, the claims are supported from the SAXS, Cryo-ET, and CD characterization.

Thank you for the supportive comment.

7. Are there other experiments or work that would strengthen the paper further?

The novelty of the paper could be improved if more phases were shown to be accessible using this technique. The researchers show that the relative size of the AuNPs influences the symmetry of the resulting lattice, studying two sizes of AuNPs that access 2D square lattices and 2D hcp lattices. What is the effect of using AuNPs larger than 12 nm? Is there another phase that can be accessed? If multiple chiral structures/phase transitions could be achieved by changing the relative size of the AuNP to the TMV, the novelty of the work would increase significantly and then this work would be appropriate for publication in Nature Communications.

We agree that showing more 2D lattice geometries is highly interesting, which we wish to further clarify here. Our results show that increasing the $D_{\text{AuNP}}/D_{\text{TMV}}$ ratio yielded a square lattice instead of the typical hexagonal lattice. The change of symmetry was explained by the fact that large AuNPs can bind to four TMVs simultaneously, whereas small AuNPs can only bind to three AuNPs simultaneously. The reviewers correctly raise the question, whether a further increase of AuNP size would allow a diversity of structures.

It could be assumed that larger AuNPs could bind to five or even six TMVs. An inspection of size ratios yield that from a particle size point-of-view a five-fold symmetry is indeed allowed even for the studied particles. However, the five-fold symmetry is not allowed for such 2D periodic structures, that is 2D lattices with five-fold symmetry is forbidden and cannot be observed. On the other hand, a six fold symmetry including six TMVs bound to each AuNP would require much larger cationic AuNPs than are synthetically available. A comparative study would require highly cationic particles. High- quality monodisperse cationic AuNPs are limited to sizes below 18 nm (Hassinen *et.al* 2015), and the size distribution of large cationic AuNPs is non-uniform, which hinders an efficient self-assembly of superlattices.

Also, if magnetic particles were used (or a mixture of magnetic and plasmonic), the packing and helicity of the structure could be changed by altering the magnetic field. Such a system, which would be responsive to external stimuli, would be quite unique and interesting.

We thank the reviewer for this excellent suggestion and we agree on the point that novelty and general interest of the manuscript is increased by demonstrating stimuli responsive functions. One of the ultimate goals of studies on self-assembly is to develop rational methods to produce stimuli-responsive functional materials.

It should be emphasized that synthesizing of cationic magnetic nanoparticles is highly challenging, and therefore a detailed study on combining different physical properties in a modular fashion in the superlattice wires would be worth an additional publication. However, to show the potential of the assembly approach presented in this manuscript we carried out a study that demonstrates that magnetic properties of the superlattice wires can be enhanced by post-functionalization of the superlattice wires (Supplementary Figs. 13 and 14). After functionalization with magnetic nanoparticles, the superlattice wires become responsive to magnetic fields and can be oriented to produce a plasmonic polarizer.

8. Questions for the Authors

A. The SAXS peaks appear to shift to lower q as the ratio of AuNP:TMV is increased (Figure 2h), indicating that the spacing in the structure is increasing. Why is this the case?

The SAXS peaks are broader for suboptimally packed structure, but the peaks are not essentially shifted. For us, such a conclusion would be an overinterpretation of the data. However, it is likely that a suboptimal packing causes a minor increase in the lattice constant, as AuNPs are increasingly crowding between TMVs.

B. Have different cationic ligands been tried?

Yes, the smaller cationic AuNPs were functionalized with a different ligand. We observed that increased flexibility of ligands results in a more compact lattice than would be expected purely on the basis of the nanoparticle size (core + ligand). The same was observed when using the soft PAMAM G5 particle. Large cationic particles with a short ligand resulted in a lattice constant that was in very good agreement with the particle sizes (no significant flexibility of particles observed).

C. What is the average length of the wires? Does this change at all with NP ratio (or other factors)?

The macroscopic habit of the structures is determined by nucleation and growth. After nucleation, the assemblies are expected to grow indefinitely, if the growth is not limited for example by lack of particles (the growth is in practice terminated when no free AuNPs or TMVs are present in the solution). The length as well as the width of the assemblies can be increased by decreasing the assembly rate (slow dialysis, less nucleation, fast growth). This was essentially demonstrated by the *in situ* optical microscopy of a fast assembly formation (Supplementary Fig. 2) that did not result in as large wires as slow dialysis. In that experiment the ionic strength decreased rapidly below the assembly/nucleation threshold. The average length is several hundred microns as observed from the optical microscopy images.

9. Clarity of the manuscript

The manuscript is clearly written and organized. See detailed comments below.

[Page 3, line 7] this is an extremely long, run on sentence

[Page 3, line 11] What does “delicate” mean in this context?

[Page 3, line 14] TMV has been used to assemble chiral NP assemblies in the past and “exquisite” is quite a subjective term. Perhaps this could be revised to a more specific comment about the improvements made.

[Page 3, line 18] The phrase “at the onset of aggregation” is unclear and could be replaced with “dispersed.”

[Page 5, line 12] “This is essentially one of the key findings” should be removed. You could say,

“Importantly/ Significantly, it shows that like-charged...” in the next sentence instead, to make the same point.

[Page 7, line 20] “continuous” is spelled incorrectly and should be changed to “continuos”

We thank the reviewer for these valuable comments that improve the clarity of the manuscript. The manuscript has been revised according to these comments.

References

1. Bromley, K. M.; Patil, A. J.; Perriman, A. W.; Stubbs, G.; Mann, S. *Journal of Materials Chemistry* 2008, 18, (40), 4796-4801.
2. Dujardin, E.; Peet, C.; Stubbs, G.; Culver, J. N.; Mann, S. *Nano Letters* 2003, 3, (3), 413-417.
3. Tseng, R. J.; Tsai, C.; Ma, L.; Ouyang, J.; Ozkan, C. S.; Yang, Y. *Nature nanotechnology* 2006, 1, (1), 72-77.
4. Fowler, C. E.; Shenton, W.; Stubbs, G.; Mann, S. *Advanced Materials* 2001, 13, (16), 1266-1269.
5. Lee, S.-Y.; Royston, E.; Culver, J. N.; Harris, M. T. *Nanotechnology* 2005, 16, (7), S435.
6. Kobayashi, M.; Tomita, S.; Sawada, K.; Shiba, K.; Yanagi, H.; Yamashita, I.; Uraoka, Y. *Optics express* 2012, 20, (22), 24856-24863.
7. Warner, M. G.; Hutchison, J. E. *Nature materials* 2003, 2, (4), 272-277.
8. Kostianen, M. A.; Hiekkataipale, P.; Laiho, A.; Lemieux, V.; Seitsonen, J.; Ruokolainen, J.; Ceci, P. *Nature Nanotechnology* 2013, 8, (1), 52-56.

Reviewer #2 (Remarks to the Author):

This article describes the co-assembly of 12 nm gold particles with tobacco mosaic virus (TMV) into higher ordered structures. Apparently, the gold particles can bridge up to four TMVs, which leads to the formation of chains of particles lying between the TMVs and thus forming ribbons of TMV-particle hybrids with micrometer diameter and hundreds of micrometer in length. By varying the respective concentrations of the the two constituents or of the ionic strength of the buffer solution, the authors reveal details of the process of formation. The growth is explained by cooperative assembly and zipper-like closure of the particles between the TMVs leading to the formation of large superlattice structures. Further, the intrinsic chirality of the TMVs is passed on to the superstructures leading to twisting of the ribbons and thus measurable CD signals of the curled gold particle chains.

While I think this is a well written manuscript describing an interesting effect, I am undecided to recommend publication in *Nature Communications* in its current state. Although the authors have characterized the emerging structures rigorously, the story would benefit from a clearer purpose. Currently it appears to be of the sort "Let's see what happens if we mix A with B". E.g. if the hypothesized effect of the components assembling in a zipper-like fashion could be observed dynamically or could be modelled in silico there might evolve more general principles. Alternatively, the authors could rewrite the conclusion and instead of mainly repeating the content of the paper elaborate more on the "future materials" they have in mind.

We thank the reviewer for this valuable comment. Indeed, to derive general rules that applies to an arbitrary combination of colloidal rods and spheres will require even more examples than those provided in the manuscript. However, our highly systematic approach to produce well-defined hierarchical structures

reveals a self-assembly mechanism that can serve as a basis for developing future materials. In order to emphasize the potential of the self-assembly process, we carried out studies that show that the formed superlattice wires can be further processed to yield multifunctional materials. The new study demonstrates that magnetic properties of the superlattice wires can be enhanced by post-functionalization of the superlattice wires (Supplementary Figs. 13 and 14). After functionalization with magnetic nanoparticles, the superlattice wires become responsive to magnetic fields and can be oriented to produce a plasmonic polarizer.

In addition, the conclusions were rewritten to take into account the potential of introducing co-operative self-assembly of superlattice wires in even industrial processes.

Minor comments:

While the scheme in Fig 1g makes sense to me (decreasing ionic strength allow aggregates to dissolve and formation of bonds with the TMVs, I have trouble understanding the statement:"Upon decreasing ionic strength, AuNPs are concentrated into the superlattice wires leading to a smaller AuNP interparticle distance." A bit more explanation of the rationale behind that statement would be helpful.

Lowering the ionic strength installs the electrostatic attraction between the AuNPs and TMV, which brings AuNPs closer to each other to form the superlattice structure. This has also been schematically presented in Figure 4b.

"... the asymmetric helical and exactly defined structure of TMV can transfer chirality into the higher order structures." into the higher order of these structures? or ... into higher ordered structures?

We have revised the section to refer non-specifically to the higher order structures.

We would sincerely like to thank both referees for their valuable comments, which have helped us to revise the manuscript, highlight the novelty and improve the overall quality of our work. We hope that our manuscript is now suitable for publication in *Nature Communications*.

Response to Reviewers' comments:

Reviewer #1 (Remarks to the Author):

The novelty of the work is better stated and defended by the introduction of magnetic nanoparticles into the system to produce films of oriented superlattice wires, which demonstrate properties of a plasmon polarizer. In addition, upon reconsideration the descriptions of the helical nature of the superlattice via the bending and torsion mechanism are unique and informative. Furthermore, the formation of well-ordered 3D chiral structures allows for improved correlation of optical properties with structure.

However, in addition to elaborating on the practical applications, it would be valuable to discuss how these findings impact the field from a scientific perspective in the conclusion. For example, how would one use the information that you have learned in this work to inform the rational design of other types of chiral materials? Or, could one envision that the bending and torsion model might be applicable to other systems? In other words—beyond it being an interesting phenomenon, what implications does this work have for the field of self-assembly? If these minor concerns can be addressed, I believe the manuscript would be suitable for publication in *Nature Communications*.

We are pleased that the reviewer is contented with our revisions and now supports the novelty of the findings. As requested, the conclusion section has been revised to include visions how the observed phenomenon might impact the research field.

Reviewer #2 (Remarks to the Author):

The authors have addressed the comments of the referees. The data of SI Figs 13 and 14 (magnetic induced polarization switching) are very impressive and I think they deserve space as a Figure 5 in the main text. Generally, the article has improved significantly and is now ready for publication.

We thank the reviewer for the positive comments that support the publication of the manuscript. Figs S13 and S14 have now been combined and moved to the manuscript as Figure 5. The text has been modified accordingly to explain the new figure.

We would sincerely like to thank both referees for re-evaluating our revised manuscript and providing final comments. We hope that our manuscript is now suitable for publication in *Nature Communications*.